# Element Rich Area Associated with Human Health Disorders: A Geomedical Science Approach to Potentially Toxic Elements Contamination

**DOI:** 10.3390/ijerph182212202

**Published:** 2021-11-20

**Authors:** Sri Manovita Pateda, Masayuki Sakakibara, Koichiro Sera

**Affiliations:** 1Medical Faculty, Universitas Negeri Gorontalo, Jenderal Sudirman Street 6, Gorontalo City 96100, Indonesia; 2Research Institute for Humanity and Nature, Kyoto 603-8047, Japan; sakaki@chikyu.ac.jp; 3Graduate School of Science and Engineering, Ehime University, Matsuyama City 790-8577, Japan; 4Cyclotron Research Center, Iwate Medical University, Takizawa 020-01673, Japan; ksera@iwate-med.ac.jp

**Keywords:** copper, mercury, geomedical science, Gorontalo

## Abstract

(1) Background: Geomedical science focuses on the relationship between environmental impact and human health. The abundance of elements in a geographic area is reflected accumulation of these elements in humans. This study aims to describe the relationship between concentrations of geologic elements and accumulations in the human body as well as element-related symptoms. (2) Methods: Geogenic sampling was conducted in an Artisanal and Small-Scale Gold Mining (ASGM) area and around residential areas in Indonesia, and samples were analyzed using particle-induced X-ray Emission (PIXE). Head hair was sampled, and health assessments were performed to determine heavy metal exposure, especially to copper and mercury. (3) Results: Results show that potentially toxic elements’ accumulation in the human body follows the abundance of these elements in the geographic area, which then affect health and manifest with specific signs and symptoms. East Tulabolo is an area rich in copper (hazard quotient (HQ) in dust = 152.8), and most of the population shows the sign of Kayser–Fleischer rings. Likewise, the Dunggilata area has the highest concentration of mercury, especially in the dust (HQ = 11.1), related to ASGM activity in residential areas. (4) Conclusions: This study concludes that the geogenic concentration of elements parallels the accumulation of human tissue and manifests with element-related signs and symptoms.

## 1. Introduction

### 1.1. Background

Human health is constantly influenced by exposure to potentially toxic elements in the environment [1,2,3,4,5,6,7,8]. Since ancient times, human interaction with potentially toxic elements has been quite close, through exposure to medicine, equipment, and other sources. Heavy metal concentrations in the environment are quite high, with contaminants in food or geogenic materials. In this industrial age, large-scale mining has caused occupational diseases in the form of poisoning by various toxic metals [1]. Potentially toxic elements do not undergo metabolism in the human body; instead, they accumulate there and combine with ligands that are biochemical, resulting in toxic effects [1,9,10,11].

Heavy metal contamination can be viewed from the perspective of geomedical science usually known as medical geology, a science that studies health problems related to “location”, based on the abundance of potentially toxic elements in the area that affects human health both directly and indirectly [2,12]. The journey of heavy metal elements into the body’s metabolic system is a complex process, influenced by many heterogeneous determinants. The essential threat of heavy metal exposure is access to the elements and their fate after entering human cell system. The basic principle of exposure to an agent is that it must enter the human body, which acts as a host, by first passing through a medium. The media of toxic elements in the geosphere and sociosphere are air, dust, food, water, and soil. Exposure to potentially toxic elements in humans occurs through inhalation, ingestion, and skin absorption [13,14,15].

The complexity of metabolic processes in the human body depends on many factors, both internal and external. Toxicokinetics and toxicodynamics of potentially toxic elements in the body vary depending on the character and history of human health. The elimination process is through two routes, primarily excretion through urine or feces and through short- or long-term accumulation pathways. Long-term retention manifests as signs of heavy metal exposure; for example, Kayser–Fleischer rings are as a sign of copper accumulation in the iris, and analysis of heavy metal concentrations in hair reveal it as a place of accumulation.

Analysis of long-term heavy metal accumulation in the human body shows that it manifests in specific signs and symptoms. Previous studies have focused on the analysis of mercury as the main pollutant in gold mining. This study focuses on two elements based on geologic character studies and sources of heavy metal pollution: copper (Cu) and mercury (Hg). The overall aim of this study is to provide information that can prove the predictability of exposure to these elements leading to accumulation in the human body.

### 1.2. Geologic Character of Gorontalo Province, Indonesia

Gorontalo Province consists of two geologic sheets: Tilamuta and Kotamobagu. The north side of Sulawesi Island, where Gorontalo Province is located, is a volcano-plutonic arc [16]. Outcrop observations show a malachite layer in the porphyry Cu oxidation zone. In the process of dismantling Cu ore from sulfide minerals, the Cu is carried away by rainwater, which is then deposited in the soil and water [16].

### 1.3. Artisanal and Small-Scale Gold Mining

Artisanal and Small-Scale Gold Mining (ASGM), as defined by the United Nations Environment Program, is gold mining conducted by individual miners or small enterprises with limited capital investment and production costs [3,17,18,19]. Processing of ore bearing gold in the ASGM area uses mercury as a gold metal binder. Initially, the effects of mine pollution affected workers and communities around the mine; however, the impact of Hg contamination can extend to the communities who live far from the mine area, and Hg contamination from mining is slowly but surely expanding.

Pateda et al., in their research on Gorontalo Province, reported a tendency for chronic Hg accumulation in human hair samples, and the chronic effect of Hg vapor on respiratory health [20]. This detrimental effect is not only experienced by the miners, but also by the people who live around the mining area even extending to a radius of a longer distance [21,22,23].

Generally, ASGM activity usually comprises the following steps: (a) Extraction, miners exploit alluvial deposits (river sediments) or hard rock deposits; (b) Processing, the gold is liberated from other minerals. Trommels are widely used and added with mercury in this process; (c) Concentration, in Gorontalo mining, the ore is then concentrated and washed with the help of sluices and pans; (d) Amalgamation, elemental mercury is used to obtain a mercury–gold alloy called an “amalgam”; (e) Burning, the amalgam is heated to vaporize the mercury and separate the gold; and (f) Refining, sponge gold is further heated to remove residual mercury and other impurities [24,25,26,27].

## 2. Materials and Methods

The research data for this study include geologic information and results of human health collected from 2018 to 2019. The geologic data comprise two types of samples: soil and dust. The health assessments were carried out on 192 respondents classified into two groups based on residence in a polluted or a non-polluted (control) area. Of the 192 participants, 108 were from polluted areas (villages of East Tulabolo, Dunggilata, Hulawa, and Bumela) and 84 were from control areas (villages of Bongo and Longalo).

The polluted area was defined as an area that is close to ASGM, whereas the control area is an area where no mining activity is present. Geologic samples were taken from several points, both in the mining area and around residential areas.

### 2.1. Soil and Dust Sample Analysis

Soil samples taken from 4 mining areas (East Tulabolo, Dunggilata, Hulawa, and Bumela) and from 2 control areas (Bongo and Longalo). Ground soil was sampled from a depth of 20–30 using a shovel and then placed into a plastic bag.

Dust sampling was conducting using a soft toothbrush, and the samples were stored in a paper bag. Each dust sample taken at a point close to the location of the ground soil sample. All soil and dust samples were analyzed using particle-induced X-ray emission (PIXE) conducted at the Cyclotron Research Center, Iwate Medical University, Japan. Sample preparation was required based on a standard method for PIXE. First, the soil and dust samples were ground to powder using a ball mill powdering system. Furthermore, the samples were mixed with the internal standard using the palladium on carbon powder method [28]. The normal concentration range of the soil and dust samples were calculated from the average values in the world [29]. The threshold value for Hg and Cu were 52 and 20 mg/kg, respectively.

### 2.2. Hair Sample Analysis

A grouping of scalp hair approximately 3 cm long was obtained from the root in the occipital region of each respondent. This approach meant that long hairs were sampled and then cut to 3 cm and only the 3 cm on the root side was analyzed. Hair samples went through several processes before being ready to send for inductively coupled plasma mass spectrometry (ICP-MS) analysis. This process was divided into three stages: stage (1) pre-washing; stage (2) washing; and stage (3) post-washing. The washing process by the ultrapure water (MilliQ^®^) aimed to remove contaminants from the hair, such as dust, dirt, bacteria, and other possible elements.

The scalp hair samples were analyzed by ICP-MS in the Research Institute for Humanity and Nature, Kyoto, Japan. Several steps for the sample and standard solution preparation were carried out. Analytical accuracy and precision were verified using Certified Reference Material No.13 (PerkinElmer, Waltham, MA, USA). Rhodium solution (Wako Pure Chemical Industries, Osaka, Japan) was used as internal standard solution.

### 2.3. Hazard Quotient Quantification

An appropriate assessment method depends on the known toxic effect of the chemicals and comprises the chemical mixture, availability of toxicity data, and quality of available exposure data [13]. The potential health hazard from exposure to each chemical is estimated by calculating its individual hazard quotient (HQ) with the following formula [13]:$$\mathrm{HQ}=\frac{Chemical\;Exposure}{Standard\;exposure}$$

### 2.4. Neurologic Assessment

Evaluation of the neurologic system was performed by examining several clinical signs, for which the most dominant neurologic sign is tremor. A tremor is a rhythmic shaking movement in one or more parts of the body. According to the Protocols for Environmental and Health Assessment of Mercury Released by Artisanal and Small-Scale Gold Miners by United Nations Industrial Development Organization, this study implemented the finger to nose test. The procedure for this test is for the participant to stand still, with their legs together, arms outstretched, and eyes closed, and then to touch their fingertip to their nose. The rating scale score range is: 0, no tremor; 1, slight to moderate (amplitude 0.5–1 cm), may be intermittent; 2, marked amplitude (1–2 cm); and 3, severe amplitude (>2 cm) [30].

### 2.5. Eye Assessment

Examination of Cu accumulation used the presence of a sign called Kayser–Fleischer rings, which are dark rings that appear to encircle the iris of the eye. Using a medical flashlight, the research team checked the participants’ eyes and could see the rings with the naked eye on the inner surface of the cornea in the Descemet membrane. The rings typically appear as a golden, brown ring in the peripheral cornea.

## 3. Results

### 3.1. Element-Rich Area

#### 3.1.1. Distribution of Elements per Area

Table 1 shows the ratio of the lowest and highest values of Cu and Hg concentrations in soil and dust samples from four polluted areas and two control areas. Most of the highest Cu and Hg concentrations were present in the dust samples with the maximum values of 3055 mg/kg in the East Tulabolo area and 577 mg/kg in the Dunggilata area. Compared with the dust concentration, the ground soil sample showed a concentration of 50% of the value of the dust sample.

The distribution of Cu and Hg concentrations compared with the limit regulation value is depicted in Figure 1 for polluted and control areas. Dust samples are unavailable for two areas, Hulawa and Bumela. Most data show that the Cu concentration is above the limit value of 20 mg/kg, except in the control region. Concentration variations were found in the Hg samples, where the highest polluted area was the Dunggilata area, as one of the active ASGMs in Gorontalo Province. A high Hg value in the control area was found in the dust sample with a median value above the limit value.

#### 3.1.2. Hazard Quotient Quantification

Hazard quotient is a comparison parameter between the exposure value with the standard allowable level. Table 2 describes the HQ of each metal element by area. The concentration of Cu in the dust showed an extreme increase compared with the increase in Hg, either in the dust or in the ground soil. In fact, the Cu concentration was markedly increased in the control region. The Hg value in the control area was also quite high in the dust sample.

#### 3.1.3. Mapping of Elements

Mapping was used to show areas based on HQs parametrically due to the large gap between the lowest and the highest values. The concentration of Cu is quite high throughout the Gorontalo area, which is represented by the research sampling areas, both in the polluted and control areas. The largest HQ for Cu is depicted in the East Tulabolo area, whereas the largest Hg HQ is in the Dunggilata area, and the smallest is in the Bumela area, an average of more than 5 of HQ. These are shown in Figure 2.

### 3.2. Sign and Symptom Distribution per Area

Public health assessments related to exposure to potentially toxic elements, especially Cu and Hg, provide data for the sign and symptoms observed and reported from the general health check-up. The presence of Kayser–Fleischer rings is a sign of chronic accumulation of Cu exposure in humans, whereas exposure to Hg is indicated by tremor as a symptom of neurologic disorders related to Hg contamination. Figure 3 describe it well.

## 4. Discussion

The uptake of elements is a process that may vary considerably depending on the complexity of the living system. Moreover, the bioavailability of elements is the main factor that influences the impact of an environmental chemical species in the human body. The bioaccumulation of elements in geologic samples is used as a consideration in risk assessment, whereas biological samples are used as bioindicators of exposure to an element including potentially toxic elements such as Cu and Hg.

A potentially toxic element that has made its way into a human body has travelled through an environmental medium [2]. The media are air, water, and soil/food, which enter the human body through three pathways: inhalation, skin absorption, and digestion. After undergoing the process of cellular metabolism, the element is then eliminated in either of two ways: excreted or accumulated. The process of excretion is through urine and feces, whereas the accumulation process occurs in several organs, including scalp hair. This accumulation is used as a bioindicator to assess the effect of exposure to a heavy metal. Once accumulated, potentially toxic elements will be trapped in organ tissues for a long time.

In the context of environmental monitoring studies, bioindicators reflect organisms—or parts of organisms or communities of organisms—that contain information on the quality of the environment—or a part of the environment [31,32]. The abundance of an element in a geographic area, be it the result of natural or anthropogenic processes, characterizes the accumulation of elements in the tissues of the human body. In line with the research conducted, the abundance of Cu levels in the Gorontalo area is due to its geologic character, which is the porphyry Cu oxidation zone.

The high concentration of Hg in the environment is from ASGM activity in polluted areas. The dust samples, representative of the presence of Hg in the air, provide evidence that Hg can be carried by the wind to more distant locations. This effect is reflected in the study results showing that the Hg concentration was also high in the control area where no ASGM activity was present. The high levels of Hg in the environment in the Dunggilata area are caused by mining work, especially mining practices that use Hg in production, such as concentration and burning of amalgam, which is carried out in residential settings that cover a wider area. Therefore, this exposure leads to quite a lot of neurologic disorder effects in this area.

The passage of elements from the environment into the human body and give effect, through complex stages. There is an intermediate analysis that must be studied previously to explain the relationship between the abundance of geological elements and their concentration in the human body. This is the limitation in this study that needs to be clarified with other more in-depth studies.

## 5. Conclusions

Accumulated elements from geogenic activities in human biological tissue can result from exposure from environmental sources. Humans take potentially toxic elements into their bodies, both consciously and unconsciously, from both natural and anthropogenic sources. This incorporation leads to the accumulation of potentially toxic elements in the tissues of the human body which then manifest as the most likely signs and symptoms to appear associated with the toxicity.

## Figures and Tables

**Figure 1 ijerph-18-12202-f001:**
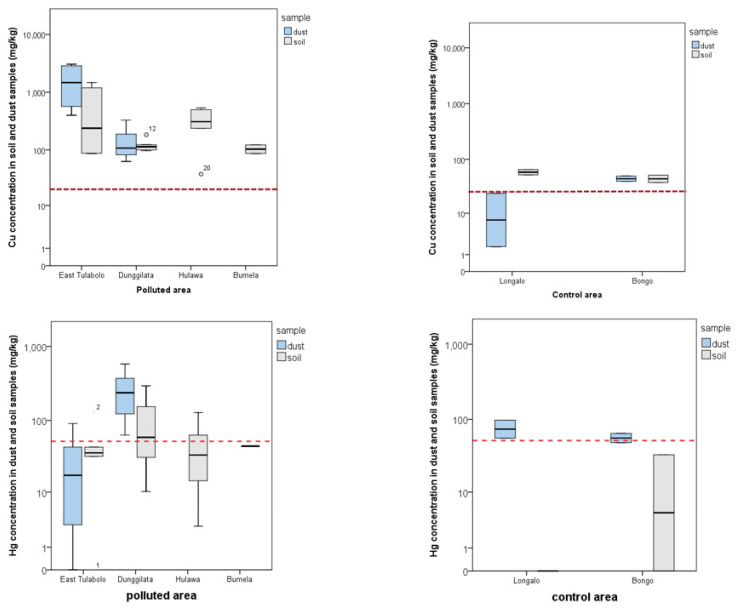
Distribution of copper (Cu) and mercury (Hg) concentrations per area based on polluted and control areas. The mean concentration value uses a logarithmic scale as there is a large difference in the lowest and highest values. The red dotted line indicates the threshold limit according to the regulation. The confidence interval of this study is 95%.

**Figure 2 ijerph-18-12202-f002:**
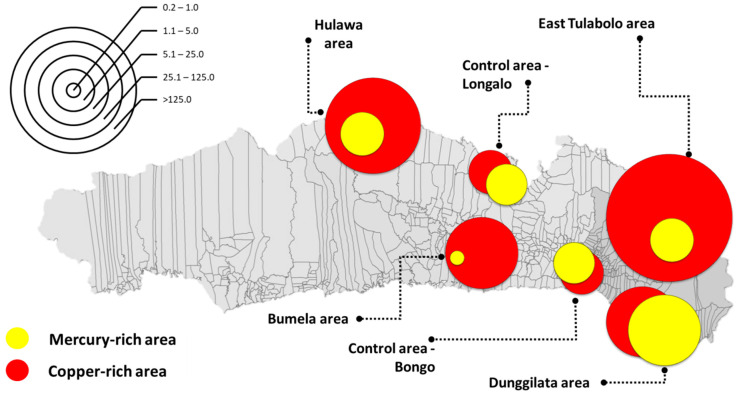
Mapping of copper (Cu) and mercury (Hg) hazard quotient per area. The scale in the left corner provides a parametric picture of the magnitude of the hazard quotient. The size of the circle is adjusted according to the hazard quotient ratio of each element, red for copper and yellow for mercury.

**Figure 3 ijerph-18-12202-f003:**
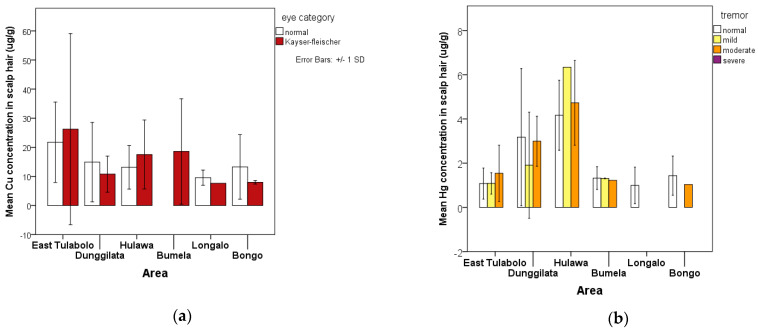
Graph depicting the relationship between the presence of health signs and symptoms related to heavy metal exposure and the accumulation of these potentially toxic elements in scalp hair: (**a**) average accumulation of copper (Cu) in human scalp hair per area shows a high number in the East Tulabolo area, and is almost as common in all areas; (**b**) average accumulation of mercury (Hg) in scalp hair per area shows a high number in the Hulawa and Dunggilata areas.

**Table 1 ijerph-18-12202-t001:** Ratio of Cu and Hg concentrations in soil and dust samples per area.

Heavy Metal in Soil	Limit Regulation (mg/kg)	Range of Potentially Toxic Elements Concentration (mg/kg) per Area (Min–Max)					
		East Tulabolo	Dunggilata	Hulawa	Bumela	Longalo	Bongo
In Soil							
Cu	20	86–**1470**	96–183	37–536	86–122	52–66	37–52
Hg	52	32–131	10–**294**	2–128	44.7	DL	33.3
In dust							
Cu	20	399–**3055**	63–329	-	-	1–24	39–50
Hg	52	15–91	63–**577**	-	-	56–98	48–65

Cu = copper; Hg = mercury; and DL = detection limit; The bold number show the highest concentration of element.

**Table 2 ijerph-18-12202-t002:** Hazard quotient of element based on area.

Area	Cu		Hg	
	Soil	Dust	Soil	Dust
East Tulabolo	**73.5**	**152.8**	2.5	1.8
Dunggilata	9.2	16.4	**5.7**	**11.1**
Hulawa	26.8	NS	2.5	NS
Bumela	6.1	NS	0.9	NS
Longalo (control)	3.3	1.2	-	1.9
Bongo (control)	2.6	2.5	0.6	1.3

Cu = copper; Hg = mercury; and NS = not sampled; The bold number show the highest HQ of element.

## Data Availability

Not applicable.

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
