# Peer review of "Element Rich Area Associated with Human Health Disorders: A Geomedical Science Approach to Potentially Toxic Elements Contamination"

_ijerph, 2021, doi:10.3390/ijerph182212202_

Round 1

Reviewer 1 Report

The authors collected environmental samples, as well as hair samples, and analyzed Cu and Hg. HQ values were calculated, and neurologic/eye assessments were conducted. However, the lack of statistical analysis makes the results less persuadable.

Major comments:

1. "heavy metal": The term "heavy metal" has been widely used for decades, but more and more criticism was raised against using this term, such as this reference below. I suggest considering a different term as appropriate through the manuscript.

Reference: Pourret, O.; Hursthouse, A. It's Time to Replace the Term "Heavy Metals" with "Potentially Toxic Elements" When Reporting Environmental Research. Int. J. Environ. Res. Public Heal. 2019, 16, 4446, doi:10.3390/ijerph16224446.

2. L97: What is the material of the shovel? Since this study focuses on metals, would a metallic shovel potentially introduce contamination? Howe the shovel was cleaned between each sampling to avoid cross-contamination. What is the material of the plastic bags? Could some plastic absorb metals?

3. L156: I think “3.055 mg/kg” should be “3,055 mg/kg”?

4. Table 1: Should “0” be expressed as less than the detection limit or N/A? Why do some values have one decimal but most of them do not. Is it related to different detection limits?

5. L195: Figure number is not corrected. Why are there no error bars here?

Minor comments:

  1. L91 – 93: I suggest the authors provide a more detailed justification for choosing the polluted and control area. Are there any references using this area as a background “blank”? How “close” is close?
  2. L99: “0f” should be “of”.

  3. L106: Not quite sure about what “threshold value” means. Is it meant “detection limit” or “reporting limit”?

  4. L114: What type of water was used for washing, RO or DI water?
  5. Figure 1: Please add replicates numbers in the legend and describe the red lines and error bar.

Reviewer 2 Report

Excellent scientific paper. The English is very good and clearly formulated, it does not need any editorial revision (this variant cannot be clicked on in the questionnaire).

However, there are the following points which colud be improved:

  1. A short chapter on how the gold is processed in the mines would be useful, as there are different chemical-physical processing methods.
  2. Explain why no water analysis have been made (population's drinking water from natural sources around the mining areas)?
  3. In the case of Cu and Hg, it would have to be stated which oxydation states was examined.

Reviewer 3 Report

Dear authors,

My  suggestion is  to avoid the term of "heavy metals", in these context please see https://doi.org/10.1351/pac200274050793 and https://doi.org/10.1007/s11631-021-00468-0 .

Taking into account that the present paper follow the content of Cu and Hg and their implication to human health, it must be highlighted from the title.

As well, the explanations of chemical symbols are not  necessary.

You can add the average quantity of the soil sample.

Overall the paper is well written and the results are clear presented.

Round 2

Reviewer 1 Report

The manuscript was improved. However, I still have a few minor comments for you to consider.

  1. Table 1: I still believe a more proper approach to express a small value, which the instrument can be detect, would be “<reporting limit” instead of “0”. If a number has to be added for some reason (e.g., statistical analysis), an alternative way to express could be half of the reporting limit.
  2. Figure 3.: I still suggest the authors adding error bars in this figure for transparency.
  3. L125: I suggest adding “by ultrapure water (Milli-Q®)”, as the authors response, in the manuscript.
